# Causal Deciphering and Inpainting in Spatio-Temporal Dynamics via Diffusion Model

**Yifan Duan**[1][*]**, Jian Zhao**[2][*][†]**, pengcheng**[5]**, Junyuan Mao**[1][*]**, Hao Wu**[1]**, Jingyu Xu**[3]**,**
**Shilong Wang**[1]**, Caoyuan Ma**[3]**, Kai Wang**[4]**, Kun Wang**[6][†]**, Xuelong Li**[2][†]

[1]University of Science and Technology of China, [2]TeleAI, China Telecom, [3]Wuhan University,
[4]National University of Singapore, [5]Beijing Forestry University, [6]Nanyang Technological University
`{duanyifan28,wslong1259,maojunyuan,wuhao2022}@mail.ustc.edu.cn,`
`{kevinxu,macaoyuan}@whu.edu.cn,pengcheng2022@bjfu.edu.cn,li@nwpu.edu.cn,`
`wk520529wjh@gmail.com,{E0823044,zhaojian90}@u.nus.edu`

## Abstract

Spatio-temporal (ST) prediction has garnered a *De facto* attention in earth sciences, such as meteorological prediction, human mobility perception. However, the scarcity of data coupled with the high expenses involved in sensor deployment results in notable data imbalances. Furthermore, models that are excessively customized and devoid of causal connections further undermine the `generalizability` and `interpretability`. To this end, we establish a framework for ST predictions from a causal perspective, termed `CaPaint`, which targets to identify causal regions in data and endow model with causal reasoning ability in a two-stage process. Going beyond this process, we build on the front door adjustment as the theoretical foundation to specifically address the sub-regions identified as non-causal in the upstream phase. By using a fine-tuned unconditional Diffusion Probabilistic Model (DDPM) as the generative prior, we in-fill the masks defined as environmental parts, offering the possibility of reliable extrapolation for potential data distributions. CaPaint overcomes the high complexity dilemma of optimal ST causal discovery models by reducing the data generation complexity from exponential to quasi-linear levels. Extensive experiments conducted on five real-world ST benchmarks demonstrate that integrating the `CaPaint` concept allows models to achieve improvements ranging from 3.7%~77.3%. Moreover, compared to traditional mainstream ST augmenters, CaPaint underscores the potential of diffusion models in ST data augmentation, offering a novel paradigm for this field. Our project is available at CaPaint.

## 1 Introduction

Deep learning methodologies have achieved groundbreaking success across a wide array of spatio-temporal (ST) dynamics systems [28, 84, 44], which include meteorological forecasting [3, 50, 59, 92], wildfire spread modeling [71, 20], intelligent transportation [29, 27, 87], and human mobility systems [27, 90], to name just a few. Traditional ST dynamics approaches, based on first-principles [4, 53], often come with high computational costs. In contrast, ST dynamic analysis methods based on deep learning are not directly reliant on the explicit expression of physical laws but are data-driven [28, 84, 3, 27], relying on training models with large-scale observable datasets [86, 65, 92].

In a parallel vein, numerous efforts aim to incorporate physical laws into deep networks [35, 8, 54, 31, 81], termed `Physics-Informed Neural Networks (PINNs)`, which blend deep learning principles with physics to address challenges in scientific computing, particularly in fluid dynamics.

---

[*]Equal contribution
[†]Corresponding authors

38th Conference on Neural Information Processing Systems (NeurIPS 2024).

`PINNs` augment traditional neural network models by including a term in the loss function that accounts for the physical laws governing fluid dynamics, such as the Navier-Stokes equations [11]. This ensures that the network's predictions are not only consistent with empirical data but also comply with the fundamental principles of fluid dynamics. However, the off-the-shelf PINNs often suffer from limited generalization capabilities, primarily due to their *customized loss function* designs and the *neglect of specific network parameter* contexts [70, 16].

To date, the data-driven deep models are still dominant in ST dynamical systems, where the numerical simulation methods and PINNs generally lag behind. The reason may stem from the rise of large models [1, 76, 28] and the high costs associated with collecting ST data from sensors [93, 39], which creates a significant conflict between the increasing size of **data-hungry** models and the **uneven, insufficient** data collection. To this end, in the ST domain, there is looming research aimed at enhancing the causality and interpretability of models.

Unfortunately, research into causality within the field of ST dynamics is lagging. Although some work has considered causal design, due to specific domain constraints and architectural design, it can only enhance the tailor-made capabilities of the model for specific tasks [95, 38]. Moreover, causal discovery tools [12, 15] applied to ST systems often confront the "curse of dimensionality" issue during dimension reduction, despite their effectiveness in elucidating causal relationships from statistical data [75, 47]. Furthermore, `NuwaDynamics` [82] for the first time proposed decomposing causal and non-causal regions in ST sequences and enhancing the robustness and generalizability of downstream model training by generating more potential distribution ST sequences through mixup [100]. CauSTG [106] and CaST [95] address the issue of ST distribution shifts by implicitly modeling the time series embeddings and employing intervention techniques to observe these shifts.

Though promising, CauSTG [106] and CaST [95] focus on modeling graph-related data, they lack an understanding of high-dimensional observational data (Dimension D < 256). `NuwaDynamics`, on the other hand, explores all environments through backdoor adjustments [51], generating a vast number of sequences, which lead to nearly $\mathcal{O}(T \times \mathcal{N}_E^{\mathcal{M}(*)})$ training complexity ($T$ represents history time step, $\mathcal{N}_E$ and $\mathcal{M}(*)$ are the number of the environmental patches and mixup, respectively).

In light of this, we propose a general causal structure plugin, termed *CaPaint*, designed to decipher causal regions in ST data without adding extra computational cost, while intervening in non-causal areas to boost the model's generalizability and interpretability. Specifically, our method employs a straightforward approach to causal deciphering, utilizing a vision transformer architecture [33] for self-supervised ST data reconstruction.

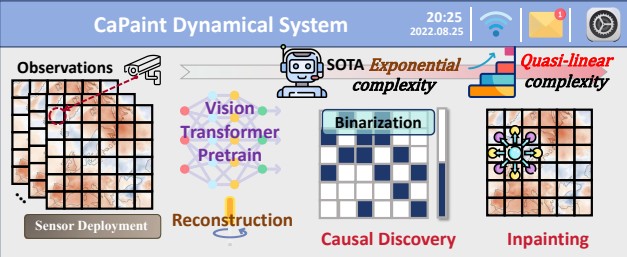

Figure 1: Illustration of the CaPaint overview and advantage across SOTA ST causal model on complexity.

During reconstruction, we leverage *attention scores* from the self-attention mechanism [23] to map onto important causal patches, thus endowing the model with interpretability. By ranking the entire set of importance scores, we define those with lower scores as environmental patches, which contribute minimally to the model. Building on this, we perform **causal interventions** in these environmental areas to aid the model in understanding more latent, complex, and imperceptible distributions, thereby enhancing the overall generalizability of the model (see Figure 1). Concretely, we mask trivial regions and perform generation using DDPM [24, 32] fine-tuned on specific ST data, which can also be interpreted as a ST data inpainting approach.

**Insight.** ❶ CaPaint obeys the causal deciphering, and guided by the principle of frontdoor adjustment [51, 52] from causal theory, CaPaint performs diffusion inpainting interventions on the environmental (non-causal) diffusion patches while reducing the temporal complexity to a manageable $\mathcal{O}(\mathcal{T} \times \mathcal{N}_{\mathcal{E}})$ (from $\mathcal{O}(T \times \mathcal{N}_E^{\mathcal{M}(*)})$ in [82]). ❷ CaPaint performs regional inpainting in a more natural manner, avoiding the predicament of repeatedly selecting and perturbing environmental patches. Through diffusion inpainting [42], it generates images that are more aligned with the global distribution. ❸ CaPaint can be understood as a ST augmenter, offering a more rational concept of ST enhancement without disrupting the inherent distribution characteristics of space and time [85]. Our major contributions can be summarized as follow:

- In this paper, we introduce a novel causal structure plugin, CaPaint, which leverages the concept of frontdoor adjustment from causal theory. CaPaint enables various backbone models to learn from a broader distribution of data while providing enhanced interpretability for the models' predictions.

- By integrating diffusion generative models with ST dynamics, CaPaint selectively perturbs non-causal regions while maintaining the integrity of core causal areas. This approach generates valuable and reliable data for scenarios where high-quality data are scarce.

- We conduct extensive experiments across five diverse and representative datasets from different domains, utilizing seven backbone models to assess the effectiveness of the CaPaint method. The empirical results demonstrate that CaPaint consistently enhances performance on all tested datasets and across all backbone models (3.7%~77.3%).

## 2   Related work & Technical Background

**Spatio-temporal Predictive Learning:** Various architectures have achieved significant predictive performance in ST domain, which can primarily be categorized as follows: CNN-based models utilize convolutional layers to effectively capture spatial features [45, 48, 77, 9]. RNN-based models, are capable of processing temporal sequence data and are well-suited for understanding temporal changes, showing excellent performance in the prediction of action continuity [69, 80, 86, 72]. GNN-based models effectively capture spatial dependencies and temporal dynamics in data, making them suitable for complex tasks involving geographic locations and temporal changes [44, 25, 36, 17, 99, 98, 83, 14]. Transformer-based models employ self-attention mechanisms to process sequential data in parallel, enhancing the learning of long-term dependencies, and have been used for ST data prediction in complex scenarios [2, 19, 89, 91, 10, 88].

**Causal inference:** causal discovery algorithms, originally devised for unstructured random vectors [66, 104], have progressively been adapted for ST data analysis [75, 47]. Within the extensive field of deep learning research, the study of causal inference aims to ensure a more stable and robust learning and reasoning paradigm. Recently, an array of techniques has been developed to delve into the nuances of causal features [60, 61, 43, 97], identifying and eliminating spurious correlations [21, 34, 56].

**Generative models** especially diffusion-based model has gained significant popularity particularly in image and video generation [24, 64, 62]. Sampling optimization algorithms have been used to accelerate the sampling process of diffusion models, significantly reducing the number of steps while improving efficiency. [67, 41]. Additionally, generative models have also been applied to 3D scene generation and point cloud processing, as demonstrated in [40, 30, 73, 74, 22, 63]

**Image Inpainting** is a technique used to fill in missing or damaged parts of an image. This field can be broadly categorized into the following types. VAE-based methods: These methods leverage Variational Autoencoders to balance diversity and reconstruction [101, 103, 26]. GAN-based methods: Since the introduction of Generative Adversarial Networks, these methods have been widely used for image inpainting [55, 102, 49]. Diffusion model-based methods: Diffusion models have recently shown outstanding performance in image inpainting [46, 68, 57].

## 3   Methodology

In this section, we systematically introduce causal structure plugin, `CaPaint`. Initially, we elucidate the methods employed in the upstream phase to delineate causal and non-causal regions (Sec 3.1). Subsequently, we showcase the theoretical underpinnings supporting the `CaPaint` (Sec 3.2). Building on this causal theory, we further engage in causal intervention within observational data (Sec 3.3). Lastly, we demonstrate how sampling-enhanced ST observations can benefit the complexity of the model's *on-device deployment* (Sec 3.4).

---

**Problem Formulation.** In ST settings, We represent ST observations as a sequence $\{X_t\}_{t=1}^{T}$, where each observation $X_t \in \mathbb{R}^{H \times W \times C_{\text{in}}}$ originates from these sequences. Our objective is to predict the trajectory for the forthcoming $K$ steps, denoted as $\{X_{t+1}\}_{t=T}^{T+K}$, with each future state $X_{t+k}$ mapped within $\mathbb{R}^{H \times W \times C_{\text{out}}}$. Here, $H$ and $W$ indicate the spatial grid dimensions, while $C_{\text{in}}$ and $C_{\text{out}}$ define the input and output dimensionality of the observations, respectively.

---

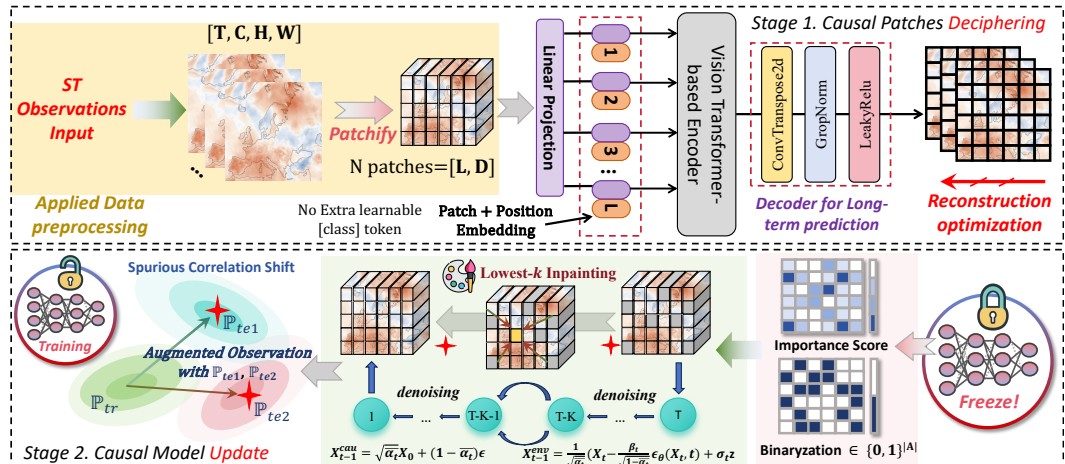

Figure 2: The details of `CaPaint`. (***Upper.***) The initial phase of discovering causal patches. (***Bottom.***) The update phase designed to eliminate spurious correlation shifts. Following the upstream training of the ViT, a diffusion model is trained in parallel. Using the identified causal patches as conditions, this generative model then performs inpainting for generating multiple sequences.

## 3.1 Causal Deciphering

To find the causal (non-causal) patches with **no labels**, we employ a self-supervised `reconstruction` approach based on the Vision Transformer (ViT) [13] to identify key regions within ST observations. ViT segments the image into multiple patches and calculates the relationships between them using a self-attention mechanism. Due to no label property, we intentionally omit the use of the `[Cls]` token in classification task and send data into ViT for encouraging *"local-to-global"* reconstruction.

Specifically, each ST data $X_t$, is divided into $N = HW/p^2$ patches, where each patch $x_t^{patch} \in \mathbb{R}^{N \times (p^2 \times C_{in})}$, with $(H, W)$ being the resolution of the original ST data and $(p, p)$ the resolution of each patch. Subsequently, each patch is mapped to a D-dimensional token through a learnable linear layer, incorporating position embedding to enhance the model's sensitivity to positional information. These tokens are then fed into successive $L$ stacked transformer blocks, as described in Equation 1:

$$
L \times \left( X' = \underbrace{X + MSA\left(LN\left(X\right)\right)}_{\text{Multi-head Attention}} \Rightarrow X_{\text{out}} = X' + \underbrace{MLP\left(LN\left(X'\right)\right)}_{\text{Residual Connection}} \right) \tag{1}
$$

where LN denotes layer normalization, and MLP represents multi-layer perceptron. The upstream self-supervised reconstruction task enables the model to learn intrinsic property of ST data. Navigating the MSA mechanism [78, 96], each patch $x_t^{patch}$ derived from the ST observation $X_t$ is transformed into queries $q$, keys $k$, and values $v$, and then calculates the relevance of each patch to others, forming a weighted representation that focuses on the most informative parts. The attention weights $A_{i,j}^h$ stored in the attention map $A^h$ in each head are computed using the scaled dot-product:

$$
\text{set } \{Q, K, V\} = X_t \psi_{tr}, \quad A^h = \text{Softmax}\left(\frac{QK^T}{\sqrt{D_h}}\right) = \begin{pmatrix} A_{1,1}^h & \cdots & A_{1,N}^h \\ \vdots & \ddots & \vdots \\ A_{N,1}^h & \cdots & A_{N,N}^h \end{pmatrix}_{A_{\{i,j\} \in 1 \to N}^h} \tag{2}
$$

where $\psi_{tr} \in \mathbb{R}^{N \times 3D_h}$ are the parameter matrices, $D_h$ represents the dimension of each head, $Q$, $K$, and $V$ collectively denote the sets of queries $q$, keys $k$, and values $v$. In our approach, the determination of causal patches, is driven by an analysis of the attention maps $A$. Each row in an attention map is normalized and represents the importance of other patches relative to the current patch $x_t^i$. However, to ascertain the overall importance of each patch across the entire input, *we aggregate the contributions by summing the values along the underline{columns} of the $A$*. To integrate insights across multiple heads, we sum these measures across all heads and then normalize the resultant vector to derive a comprehensive importance score for each patch:

$$S \in \mathbb{R}^N = \text{Softmax}\left(\sum_{h=1}^{H}\sum_{i=1}^{N} A_{i,j}^h\right) \tag{3}$$

where $S$ represents the normalized importance score vector, $A_{i,j}^h \in A$ denotes the attention that $x_t^i$ pays to $x_t^j$ for each head, $H$ is the number of heads. We sort the importance scores in $S$ and select the patches corresponding to the lowest $K$ scores as environmental patches storing in $O_e$. The remaining patches are considered causal patches $O_c$:

$$O_c = \text{Topk}\left(\lceil \mathcal{C}(S) \times \epsilon\% \rceil, \ \underset{S_i \in S}{\arg\max}\{\text{set}\left(\Psi\left(X_t\right)\right)\}\right) \tag{4}$$

where $\mathcal{C}(S)$ is the counting function, $\epsilon$ represents the proportion of patches selected as causal, and $\Psi(X_t)$ denotes the set of patches in the ST observation $X_t$. We identify the causal patches by locating the indices with the highest values in $S$ and define the non-causal parts as the environmental parts. Our goal is to perform causal interventions on the environmental parts.

### 3.2 Backdoor Adjustment v.s Frontdoor Adjustment

To address issues of ST data scarcity and poor transferability, we examine the evaluation process using a Structural Causal Model (SCM) [52], as shown in Fig 3. We represent abstract data variables by nodes, with directed links symbolizing causality. The SCM illustrates the interaction among variables through a graphical definition of causation, demonstrating the interconnected nature of these elements. As depicted in the left part, `NuwaDynamics` employs the backdoor adjustment to enhance the model's generalization performance:

➡ $\mathcal{X_C} \leftarrow \mathcal{X} \rightarrow \mathcal{X_{\backslash C}}$. The input $\mathcal{X}$ consists of two disjoint parts $\mathcal{X_C}$ (causal part) and $\mathcal{X_{\backslash C}}$ (environmental or trivial part).

➡ $\mathcal{X_C} \rightarrow \mathcal{Y} \leftarrow \mathcal{X_{\backslash C}}$. Here, $\mathcal{X_C}$ represents the sole endogenous parent that determines the ground truth $\mathcal{Y}$. However, in practical scenarios, $\mathcal{X_{\backslash C}}$ is also employed in predicting $\mathcal{Y}$, which leads to the formation of spurious associations.

In general, a model $\mathcal{F}_\emptyset$ trained using Empirical Risk Minimization (ERM) often struggles to generalize to the test data $\mathcal{D}_{te} \sim \mathbb{P}_{te}$. Such distribution shifts are often induced by variations in environmental patches. Hence, addressing the confounding effect caused by the environmental confounder is crucial. Backdoor adjustment techniques are employed to perturb the environmental components, thereby enhancing the model's potential to observe a broader range of latent distributions by forcibly perturbing the environmental variables $\mathcal{X_{\backslash C}}$ (referred to as the **do-calculus** [51] operator). Unfortunately, ❶ traversing all environmental variables is quite

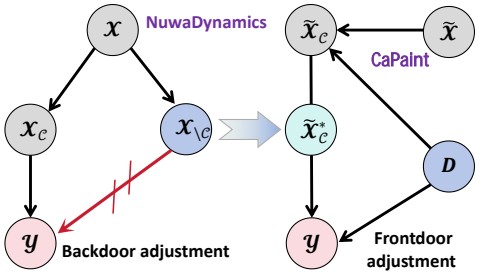

Figure 3: Different SCM architectures of `SOTA` and CaPaint.

challenging. Although `NuwaDynamics` uses Gaussian sampling to mitigate the issue of complexity, controlling Gaussian sampling in temporal sequence operations is particularly difficult. It requires meticulous adjustment of mean and variance to ensure a balance between the number of environmental samples and the training burden. ❷ Worse still, by traversing all environments, it likely violates underlying properties, including distribution shift content and nonexistent scenarios [94]. To address this issue, we employ front-door adjustment, as illustrated in the right half of the Fig 3:

• $\tilde{\mathcal{X}}_C \leftarrow \mathcal{D} \rightarrow \mathcal{Y}$. In this structure, $\mathcal{D}$ serves as a confounder, creating a misleading path between $\tilde{\mathcal{X}}_C$ and $\mathcal{Y}$. Here, $\tilde{\mathcal{X}}_C$ represents the causal component within $\tilde{\mathcal{X}}$.

• $\tilde{\mathcal{X}}_C \rightarrow \tilde{\mathcal{X}}_C^* \rightarrow \mathcal{Y}$. $\tilde{\mathcal{X}}_C^*$ acts as the surrogate variable of $\tilde{\mathcal{X}}_C$ and completes $\tilde{\mathcal{X}}_C$ to align it with the data distribution. Initially, it derives from and encompasses $\tilde{\mathcal{X}}_C$. Specifically, it envisions the potential complete observations that should exist when observing the sub-counterpart $\tilde{\mathcal{X}}_C$. Additionally, $\tilde{\mathcal{X}}_C^*$ adheres to the data distribution and upholds the intrinsic knowledge of graph properties, thus eliminating any link between $\mathcal{D}$ and $\tilde{\mathcal{X}}_C^*$. Consequently, $\tilde{\mathcal{X}}_C^*$ is well-suited to act as the mediator, which in turn influences the model's predictions ($\rightarrow \mathcal{Y}$).

In our front-door adjustment framework, we utilize **do-calculus** on the variable $\tilde{\mathcal{X}}_{\mathcal{C}}$ to eliminate the spurious correlations introduced by $\mathcal{D} \to \mathcal{Y}$. Specifically, we achieve this by summing over potential surrogate observations $\tilde{X}_{\mathcal{C}}^*$. This approach allows us to connect two identifiable partial effects: $\tilde{\mathcal{X}}_{\mathcal{C}} \to \tilde{\mathcal{X}}_{\mathcal{C}}^*$ and $\tilde{\mathcal{X}}_{\mathcal{C}}^* \to \mathcal{Y}$:

$$
\begin{aligned}
P\left(\mathcal{Y}|do\left(\tilde{\mathcal{X}}_{\mathcal{C}} = \tilde{X}_{\mathcal{C}}\right)\right) &= \sum_{\tilde{X}_{\mathcal{C}}^*} P\left(\mathcal{Y}|do\left(\tilde{\mathcal{X}}_{\mathcal{C}}^* = \tilde{X}_{\mathcal{C}}^*\right)\right) P\left(\tilde{\mathcal{X}}_{\mathcal{C}}^* = \tilde{X}_{\mathcal{C}}^*|do\left(\tilde{\mathcal{X}}_{\mathcal{C}} = \tilde{X}_{\mathcal{C}}\right)\right) \\
&= \sum_{\tilde{X}_{\mathcal{C}}^*} \sum_{\tilde{X}_{\mathcal{C}}'} P\left(\mathcal{Y}\Big|\tilde{\mathcal{X}}_{\mathcal{C}}^* = \tilde{X}_{\mathcal{C}}^*; \tilde{\mathcal{X}}_{\mathcal{C}} = \tilde{X}_{\mathcal{C}}'\right) P\left(\tilde{\mathcal{X}}_{\mathcal{C}} = \tilde{X}_{\mathcal{C}}'\right) P\left(\tilde{\mathcal{X}}_{\mathcal{C}}^* = \tilde{X}_{\mathcal{C}}^*\Big|do\left(\tilde{\mathcal{X}}_{\mathcal{C}} = \tilde{X}_{\mathcal{C}}\right)\right) \\
&= \sum_{\tilde{X}_{\mathcal{C}}^*} \sum_{\tilde{X}_{\mathcal{C}}'} P\left(\mathcal{Y}\Big|\tilde{\mathcal{X}}_{\mathcal{C}}^* = \tilde{X}_{\mathcal{C}}^*; \tilde{\mathcal{X}}_{\mathcal{C}} = \tilde{X}_{\mathcal{C}}'\right) P\left(\tilde{\mathcal{X}}_{\mathcal{C}} = \tilde{X}_{\mathcal{C}}'\right) P\left(\tilde{\mathcal{X}}_{\mathcal{C}}^* = \tilde{X}_{\mathcal{C}}^*\Big|\tilde{\mathcal{X}}_{\mathcal{C}} = \tilde{X}_{\mathcal{C}}\right)
\end{aligned}
\tag{5}
$$

$P\left(\tilde{\mathcal{X}}_{\mathcal{C}}^*|do\left(\tilde{\mathcal{X}}_{\mathcal{C}} = \tilde{X}_{\mathcal{C}}\right)\right) = P\left(\tilde{\mathcal{X}}_{\mathcal{C}}^*|\tilde{\mathcal{X}}_{\mathcal{C}} = \tilde{X}_{\mathcal{C}}\right)$ holds as $\tilde{\mathcal{X}}_{\mathcal{C}}$ is the only parent of $\tilde{\mathcal{X}}_{\mathcal{C}}^*$. With data pair $(\tilde{\mathcal{X}}_{\mathcal{C}}, \tilde{\mathcal{X}}_{\mathcal{C}}^*)$, we can feeding the surrogate observations $\tilde{\mathcal{X}}_{\mathcal{C}}^*$ into our ST framework, conditional on the $\tilde{\mathcal{X}}_{\mathcal{C}}$, to estimate $P\left(\mathcal{Y}\Big|\tilde{\mathcal{X}}_{\mathcal{C}}^* = \tilde{X}_{\mathcal{C}}^*; \tilde{\mathcal{X}}_{\mathcal{C}} = \tilde{X}_{\mathcal{C}}'\right)$. Compared to previous work `NuwaDynamics`, CaPaint utilizes causal regions to generate global surrogate variables in a more rational manner, circumventing the cumbersome need to traverse environmental variables inherent in backdoor adjustments. **In fact, backdoor adjustments often likely violate underlying properties, leading to the generation of non-existent data distributions.** The broader scenarios of CaPaint will be detailed in Appendix C.

### 3.3 Causal Intervention via Diffusion Inpainting

Building on the principles of causal analysis outlined above, we proceed to perform interventions on the environmental patches using diffusion inpainting, which enables us to manipulate the environmental areas. Initially, given the unique complexities of ST datasets, we *fine-tune* the diffusion parameters to adapt seamlessly to the domain-specific challenges, which enhances the accuracy of our interventions on environmental patches. Diffusion models learn the distribution of data through a forward noise addition process and a reverse denoising process:

$$
q(X_t \mid X_{t-1}) = \mathcal{N}(X_t; \sqrt{1 - \beta_t} X_{t-1}, \beta_t I), \quad p_\theta(X_{t-1} \mid x_t) = \mathcal{N}(X_{t-1}; \mu_\theta(X_t, t), \Sigma_\theta(X_t, t))
\tag{6}
$$

where $X_t$ represents the data state at time step $t$, undergoing a transformation from its previous state $x_{t-1}$, $\beta_t$ controls the variance of the noise added at each step in the forward process, $\mu_\theta$ and $\Sigma_\theta$ are neural network outputs that approximate the mean and covariance, respectively. The fine-tuning objective of the diffusion process is designed to approximate the data distribution more accurately. Specifically, the training objective for diffusion models, denoted as $\epsilon_\theta$, which predicts the noise, is typically defined as a simplified version of the variational bound:

$$
L_{\text{simple}} = \mathbb{E}_{X_0, \boldsymbol{\epsilon} \sim \mathcal{N}(\mathbf{0}, \mathbf{I}), \boldsymbol{c}, t} \| \boldsymbol{\epsilon} - \boldsymbol{\epsilon}_\theta(X_t, \boldsymbol{c}, t) \|^2
\tag{7}
$$

where $c$ is the condition information. In this paper, we perform inpainting on the environmental patches of ST data. Inspired by [42], we generate a mask image for each ST data where the causal patches are black and the environmental patches are white. By independently sampling the causal and environmental patches and applying the diffusion inpainting process, we are able to generate augmented ST observation data. The detailed algorithmic process is shown in Appendix A.

$$
X_{t-1}^{cau} = \sqrt{\bar{\alpha}_t} X_0 + (1 - \bar{\alpha}_t)\epsilon, \quad X_{t-1}^{env} = \frac{1}{\sqrt{\alpha_t}}\left(X_t - \frac{\beta_t}{\sqrt{1 - \bar{\alpha}_t}} \epsilon_\theta(X_t, t) + \sigma_t z\right)
\tag{8}
$$

$$
X_{t-1} = m \odot X_{t-1}^{cau} + (1 - m) \odot X_{t-1}^{env}
\tag{9}
$$

where $X^{cau}$ and $X^{env}$ denote causal patches and environmental patches, $m$ is a binary mask matrix, $\alpha_t$ represents the scaling factor at each diffusion step, determining the variance retained in the transition from $X_{t-1}$ to $X_t$. The cumulative product $\bar{\alpha}_t = \prod_{i=1}^{t} \alpha_i$ represents the accumulated scaling effect from the $T = 0$ to step $t$. Equation 9 illustrates the merging of environmental patches and causal patches. Finally, the enhanced ST observation data are stored within our temporal sequence repository to bolster the downstream backbone.

## 3.4 ST Sequence Sampling Modeling

Previous work [82] assumed that the closer the time point is to the present, the greater its influence, and thus used Gaussian sampling to select more ST data closer to the current time point. However, we argue that uniform sampling can better enhance the model's generalization ability. To enhance computational efficiency while ensuring prediction accuracy, we employ a ST sequence modeling approach that samples at each time point with a fixed probability controlled by the hyperparameter $p$. This method allows us to sample from both original and generated data at each time point, thereby creating a new spatiotemporal sequence. We use two hyperparameters: $p$, which controls the sampling probability, and $r$, which determines the number of generated spatiotemporal sequences, achieving an optimal balance between computational efficiency and prediction accuracy. The specific sampling process can be represented by the following equation:

$$X'_t = \text{Sample}(X_t, p, r) \tag{10}$$

where $X_t$ represents the collection of original and generated data at time point $t$, and $\text{Sample}(X_t, p)$ denotes the dataset obtained by sampling from $X_t$ with probability $p$. The hyperparameter $p$ is directly set as the sampling probability, while $r$ is used to specify the number of generated ST sequences.

## 4 Experiments

In this section, we will validate the effectiveness of our proposed causal structure plugin, CaPaint. We design four research questions (RQs) to comprehensively evaluate the performance of CaPaint: **RQ1:** Does CaPaint effectively enhance model performance and applicability? **RQ2:** How does CaPaint perform in data-scarce scenarios? **RQ3:** How does the performance of CaPaint compare with other augmentation methods? **RQ4:** Is CaPaint effective for long-term time step predictions? Through these research questions, we aim to validate the effectiveness and advantages of CaPaint in handling ST data from multiple perspectives.

### 4.1 Experimental settings

**Datasets.** We extensively evaluate our proposal using a diverse range of benchmark datasets spanning multiple fields, include FireSys [7], SEVIR [79], Diffusion reaction system (DRS) [6], KTH [58] and TaxiBJ+ [37]. Specifically, FireSys represents fire dynamics, SEVIR covers meteorological events, DRS involves physical control systems, KTH focuses on human motion dynamics, and TaxiBJ+ is a transportation dataset. Detailed information can be found in the Appendix B.

**Backbones and Metrics** To validate the generalizability of CaPaint, we select multiple model frameworks for our experiments, including the classic model like ConvLSTM [65], PredRNN-V2 [86], Vision Transformer (ViT) [13], MAU [5], the efficiency-focused SimVP [18], and some of the latest models like MmvP [105] and Earthfarsser [92]. Our evaluation metrics include mean absolute error (MAE), mean squared error (MSE), and structural similarity index measure (SSIM). Detailed information can be found in the Appendix D.

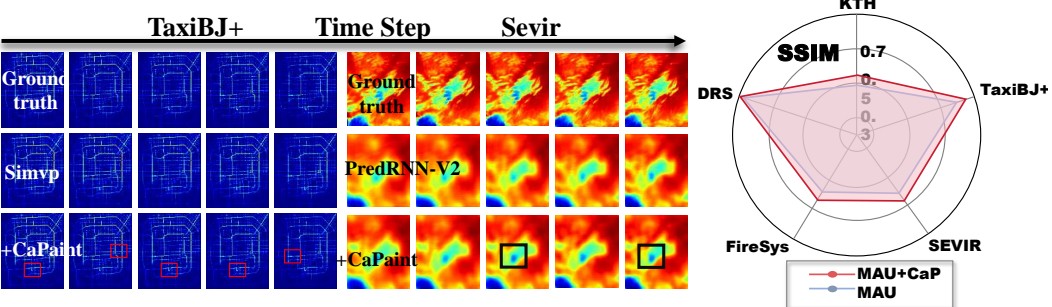

Figure 4: Visualization of prediction results for TaxiBJ+ and SEVIR datasets. The left side shows the predicted results of the last 5 frames for TaxiBJ+. The middle presents the results of long-term predictions for SEVIR, displaying the last five frames from step 10 → step 20. The right side compares SSIM metrics with and without the incorporation of CaPaint.

Table 1: This table showcases the results (five runs) differences between using the CaPaint concept (+CaP) and not using it (Ori) across various datasets. All MAE and MSE values are multiplied by 100. Blue and Red backgrounds indicate the percentage improvement (reduction) in MAE and MSE, respectively.

| Backbone (10 → 10) | Metric | TaxiBJ+ | | KTH | | SEVIR | | DRS | | FireSys | |
|---|---|---|---|---|---|---|---|---|---|---|---|
| | | Ori | +CaP | Ori | +CaP | Ori | +CaP | Ori | +CaP | Ori | +CaP |
| ViT [13] | MAE | 16.59 | 14.54 | 32.03 | 29.52 | 18.69 | 17.56 | 13.59 | 7.52 | 17.32 | 15.97 |
| | MSE | 11.40 | 8.89 | 36.11 | 32.79 | 9.93 | 9.16 | 6.21 | 1.41 | 23.40 | 21.06 |
| | Δ | 12.4% ↑ | 22.1% ↑ | 7.8% ↑ | 9.2% ↑ | 6.1% ↑ | 7.7% ↑ | 44.7% ↑ | 77.3% ↑ | 7.8% ↑ | 10.1% ↑ |
| Earthfarsser [92] | MAE | 14.57 | 12.75 | 23.56 | 20.59 | 15.23 | 14.47 | 2.03 | 1.44 | 17.15 | 16.29 |
| | MSE | 9.94 | 7.83 | 16.84 | 14.07 | 6.75 | 6.01 | 4.09 | 2.24 | 23.37 | 21.94 |
| | Δ | 12.5% ↑ | 21.2% ↑ | 12.6% ↑ | 16.4% ↑ | 5.0% ↑ | 10.9% ↑ | 29.1% ↑ | 37.8% ↑ | 5.1% ↑ | 6.1% ↑ |
| Mmvp [105] | MAE | 17.41 | 16.17 | 30.62 | 27.57 | 20.67 | 17.21 | 15.05 | 11.02 | 19.37 | 18.16 |
| | MSE | 14.22 | 12.29 | 27.31 | 22.37 | 8.45 | 7.26 | 4.11 | 2.32 | 26.09 | 24.97 |
| | Δ | 7.1% ↑ | 13.6% ↑ | 10.0% ↑ | 18.1% ↑ | 16.7% ↑ | 14.1% ↑ | 26.8% ↑ | 43.6% ↑ | 6.2% ↑ | 4.3% ↑ |
| ConvLSTM [65] | MAE | 18.22 | 16.21 | 22.77 | 20.03 | 20.51 | 18.41 | 5.43 | 3.89 | 22.22 | 10.08 |
| | MSE | 16.79 | 14.67 | 27.37 | 25.15 | 12.12 | 11.41 | 0.64 | 0.31 | 28.64 | 26.44 |
| | Δ | 13.4% ↑ | 12.6% ↑ | 12.1% ↑ | 8.1% ↑ | 10.2% ↑ | 5.9% ↑ | 28.3% ↑ | 51.6% ↑ | 9.6% ↑ | 7.6% ↑ |
| PredRNN-V2 [86] | MAE | 14.18 | 13.05 | 26.73 | 23.64 | 17.94 | 16.26 | 8.76 | 7.98 | 18.26 | 16.14 |
| | MSE | 9.60 | 7.89 | 21.45 | 19.11 | 8.54 | 7.73 | 4.37 | 4.18 | 24.71 | 23.12 |
| | Δ | 8.0% ↑ | 16.6% ↑ | 11.6% ↑ | 10.9% ↑ | 9.3% ↑ | 9.4% ↑ | 8.9% ↑ | 4.3% ↑ | 11.6% ↑ | 6.5% ↑ |
| MAU [5] | MAE | 23.28 | 20.96 | 29.54 | 27.82 | 25.07 | 24.14 | 11.84 | 9.97 | 20.67 | 18.65 |
| | MSE | 20.46 | 16.60 | 30.19 | 27.84 | 15.43 | 14.34 | 5.28 | 4.66 | 30.89 | 28.91 |
| | Δ | 10.0% ↑ | 18.9% ↑ | 5.9% ↑ | 7.8% ↑ | 3.7% ↑ | 7.1% ↑ | 15.8% ↑ | 11.8% ↑ | 9.8% ↑ | 6.4% ↑ |
| SimVP [18] | MAE | 15.91 | 13.45 | 23.21 | 20.56 | 15.48 | 14.63 | 2.12 | 1.57 | 17.01 | 15.79 |
| | MSE | 10.96 | 8.21 | 16.46 | 13.91 | 6.82 | 6.21 | 9.54 | 5.03 | 23.34 | 22.11 |
| | Δ | 15.4% ↑ | 25.1% ↑ | 11.4% ↑ | 15.3% ↑ | 5.5% ↑ | 8.9% ↑ | 25.9% ↑ | 47.3% ↑ | 8.4% ↑ | 5.3% ↑ |

## 4.2 Evaluating the Efficacy of CaPaint (RQ1 & RQ4)

In this section, we conduct extensive experiments to demonstrate the effectiveness of the *CaPaint* method. For Transformer architectures, we can directly transfer the model parameters trained in upstream tasks, thereby achieving efficient downstream training. For non-Transformer architectures, we focus on transferring the data itself to train the downstream models. The data presented in the Tab 1 show the performance improvements achieved by generating **only one** single generalized copy for each ST sequence. As shown in the Table 1, we can list the **Obs**ervations:

**Obs 1. +CaPaint consistently leads w/o Capaint settings across all datasets.** As shown in Table 1 and the right side of Fig 4, we can easily observe that introducing +CaPaint significantly improves model performance on MAE, MSE and SSIM metrics across all datasets. For example, with the ViT model on TaxiBJ+, MAE drops from 16.59 → 14.54, MSE from 11.40 → 8.89; On Diffusion Reaction Systems, MAE significantly decreases from 13.59 → 7.52, MSE from 6.21 → 1.41. This shows CaPaint's effectiveness in boosting performance in various domains.

**Obs 2. +CaPaint enhances model local insights ST scenarios.** By analyzing the left side of Figure 4, we clearly see that the +CaPaint effectively reduces the model's prediction loss. Moreover, it is observed that +CaPaint provides more accurate predictions in finer details, closely aligning with the actual result curves. This demonstrates CaPaint's capability to enhance prediction accuracy and reliability, ensuring that the forecasts closely mirror real-world outcomes.

**Obs 3. +CaPaint remains effective in long-Term ST predictions.** By analyzing the middle of Figure 4, we observe that +CaPaint continues to demonstrate its effectiveness in long-term time step predictions for ST tasks. For instance, the details in the SEVIR dataset predictions improve significantly, indicating that CaPaint is still applicable and beneficial in challenging ST tasks.

## 4.3 Performance in Data-Scarce Scenarios (RQ2)

To assess the performance of *CaPaint* in data-scarce scenarios, we conducted experiments using **varying proportions of training data** across multiple datasets and backbones. Specifically, we measured the SSIM improvement at different training data proportions, demonstrating the generalizability and robustness of *CaPaint*.

**Obs 1. CaPaint shows consistent improvements across all training data proportions.** As shown in Figures 5 and 6, *CaPaint* consistently improves SSIM across all

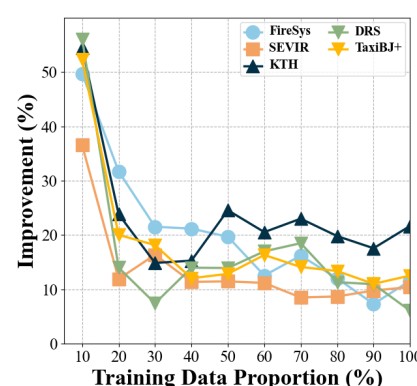

Figure 5: SSIM improvement across different datasets using the Mmvp model

training data proportions. This indicates that CaPaint is effective regardless of the amount of training data available, reinforcing its versatility and applicability in diverse scenarios.

**Obs 2. Significant performance gains in low data scenarios.** The results indicate that CaPaint yields substantial performance improvements, especially in low data scenarios. For instance, with only 10% of the training data, the SSIM improvement is most pronounced, highlighting the method's effectiveness in data-scarce environments. For example, in the TaxiBJ+ dataset with ViT backbone, the SSIM improvement reaches up to more than 50%, showcasing CaPaint's capability to enhance model performance with limited data.

**Obs 3. Diminishing returns with increased training data.** While *CaPaint* consistently enhances performance, the degree of improvement diminishes as the proportion of training data increases. This trend suggests that the primary benefits of *CaPaint* are most evident when data is scarce, but the method remains beneficial even as more data becomes available.

**Obs 4. CaPaint demonstrates superior performance with equivalent data volumes.** As illustrated in Fig 7, when comparing 25% original plus 25% augmented data with 50% original data, *CaPaint* achieves lower MAE and MSE. This demonstrates that *CaPaint* consistently outperforms the original model by effectively using a mix of original and augmented data, which together match the data volume used by the original model alone.

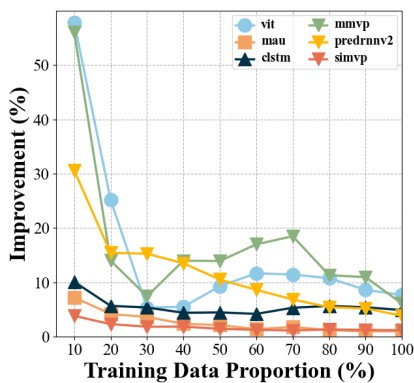

Figure 6: SSIM Improvement on DRS across various backbones

### 4.4 Performance Comparison (RQ3)

Table 2: Comparison between CaPaint and other data augmentation methods across various datasets.

| Datasets | Flip | Rotate | Crop | NuWa | CaPaint |
|---|---|---|---|---|---|
| DRS | $2.10_{\pm 0.16}$ | $2.11_{\pm 0.19}$ | $2.34_{\pm 0.26}$ | $2.02_{\pm 0.09}$ | $1.57_{\pm 0.14}$ |
| KTH | $23.15_{\pm 1.95}$ | $23.14_{\pm 1.67}$ | $23.11_{\pm 1.83}$ | $22.32_{\pm 0.94}$ | $20.56_{\pm 1.02}$ |
| SEVIR | $15.41_{\pm 1.49}$ | $15.45_{\pm 1.32}$ | $15.95_{\pm 1.64}$ | $15.14_{\pm 1.57}$ | $14.63_{\pm 1.89}$ |
| TaxiBJ+ | $16.47_{\pm 0.99}$ | $16.39_{\pm 1.32}$ | $15.94_{\pm 1.45}$ | $15.11_{\pm 0.87}$ | $12.87_{\pm 0.76}$ |
| FireSys | $17.02_{\pm 2.17}$ | $17.07_{\pm 1.94}$ | $17.15_{\pm 2.45}$ | $16.68_{\pm 1.79}$ | $15.79_{\pm 1.88}$ |

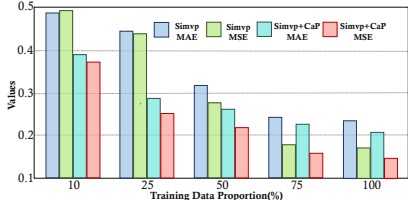

Figure 7: Visualizations in both MAE and MSE with Simvp and + CaP at various training data proportions.

In this section, we compare the performance of different data augmentation methods. Tab 2 shows the model performance using various data augmentation methods across multiple datasets, measured by MAE. It can be seen that traditional data augmentation methods, such as flipping, rotation, and cropping, produce results that are either on par with or slightly worse than the original data. Take the FireSys dataset as an example, MAE increased from $17.01 \rightarrow 17.07$ after rotation augmentation. This indicates that conventional data augmentation methods may **disrupt the intrinsic properties** of ST data, thereby negatively impacting model performance.

In contrast, our method *CaPaint* achieves the best performance **across all datasets**. For instance, on the TaxiBJ+ dataset, the MAE with *CaPaint* augmentation is 12.87, which is significantly better than the MAE of 15.11 with *NuwaDynamics* manual mixup augmentation and the MAE of 15.94 with other traditional augmentation methods such as cropping. These results highlight the advantage of our method in preserving the integrity of ST data properties. CaPaint not only effectively avoids the disruption caused by data augmentation processes on ST data characteristics but also significantly enhances the model's predictive capability.

## 5 Conclusion & Future Work

In this study, we advance the exploration of applying front-door adjustment and causality principles to spatio-temporal forecasting tasks through the introduction of *CaPaint*. Building upon the foundation of upstream self-supervised learning, we identify causal regions as crucial elements for generating

comprehensive and potential data distributions. By integrating diffusion generative models, we ensure the generated data's rationality and generalizability, thereby enhancing the downstream models' ability to generalize beyond the observed distribution and improving their interpretability. Moving forward, we plan to explore various generative models for the production of arbitrary-channel ST data to enhance the *CaPaint* robustness.

## 6   Acknowledgement

This work was supported by National Natural Science Foundation of China (62476224).

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

# A    CaPaint Inpainting Algorithm

---

**Algorithm 1** Causal Intervention with Diffusion Inpainting

---
1: **Input:** ST observation data $X$, masked image $X_{mask}$
2: **Output:** Augmentation ST observation dataset $X_A$
3: Initialize $X_T \sim \mathcal{N}(0, I)$ where $T$ is the total number of diffusion steps
4: /* Iterate backwards through diffusion steps */
5: **for** $t = T$ **to** 1 **do**
6:     /* Sample Gaussian noise $\epsilon$ */
7:     $\epsilon \sim \mathcal{N}(0, I)$
8:     /* Sample causal region */
9:     $X_{t-1}^{cau} = \sqrt{\bar{\alpha}_t} X_0 + (1 - \bar{\alpha}_t)\epsilon$
10:    /* Sample Gaussian noise $\mathcal{N}$ */
11:    $z \sim \mathcal{N}(0, I)$
12:    /* Causal Intervention on Environmental Patches */
13:    $X_{t-1}^{env} = \frac{1}{\sqrt{\alpha_t}} \left( X_t - \frac{\beta_t}{\sqrt{1-\bar{\alpha}_t}} \epsilon_\theta(X_t, t) + \sigma_t z \right)$
14:    /* Combine causal and environmental patches */
15:    $X_{t-1} = m \odot X_{t-1}^{cau} + (1 - m) \odot X_{t-1}^{env}$
16:    $X_t \sim \mathcal{N}(\sqrt{1 - \beta_{t-1}} X_{t-1}, \beta_{t-1} I)$
17: **end for**
18: **return** $X_A$ as the augmentation dataset

---

The algorithm for **Causal Intervention with Diffusion Inpainting** aims to augment ST observation data through a series of diffusion steps that iteratively refine the data by applying causal interventions and combining them with environmental patches. Here is a detailed step-by-step description:

- **Input:** The original ST observation data $X$, and a masked image $X_{\text{mask}}$.

- **Output:** An augmented ST observation dataset $X_A$.

- The process begins by initializing $X_T$, which represents the data at the final diffusion step, to be a sample from a normal distribution centered at zero with identity covariance.

- The main loop of the algorithm runs backward from the last diffusion step $T$ to the first. In each step:

  1. Gaussian noise $\epsilon_t$ is sampled to simulate the diffusion process.
  2. A causal region $X_{\text{cau}}$ is sampled where the causal effect is calculated as a blend of the original data and the Gaussian noise, emphasizing areas of interest that should retain more original data characteristics.
  3. Gaussian noise $N_t$ is sampled again, providing variability to the non-causal or environmental regions.
  4. The environmental patches $X_{\text{env}}$ are updated using the data from the previous step adjusted by a damping factor and the added noise, simulating environmental changes.
  5. The causal and environmental patches are then combined, where the mask $M$ determines the specific locations for the causal and environmental updates in the data, specifying which parts are from the causal region and which are from the environmental region.
  6. The data for the next step, $X_{t-1}$, is computed by normalizing the combined updates, preparing it for the next iteration or output if it is the first step.

- Finally, the algorithm outputs the augmented data set $X_A$, which is the result of the iterative causal intervention and environmental blending over the diffusion process.

# B    Details of experiments

**SSIM** stands for Structural Similarity Index Measure, which is a method for measuring the similarity between two images. It compares the structural information of the images, including luminance,

contrast, and texture, to determine how similar they are. SSIM is commonly used in image and video processing applications, such as image compression and quality assessment.

**PSNR** stands for Peak Signal-to-Noise Ratio. It is a measure of video or image quality that compares the original signal to the compressed or transmitted signal. The higher the PSNR value, the better the quality of the compressed or transmitted signal. PSNR is commonly used in video and image compression applications to evaluate the effectiveness of compression algorithms.

**MSE (Mean Squared Error)** loss is a commonly used loss function in machine learning and deep learning models. This loss function calculates the average of the squared differences between the predicted and actual values.

**Datasets.** Here we summarize the details (Tab. 1) of the datasets used in this paper:

- TaxiBJ+: This dataset contains trajectory data obtained from the GPS of taxis in Beijing, divided into two separate channels: inflow and outflow. Additionally, the dataset has been extended from 32×32 to 128×128 by collecting recent trajectory data from Beijing.
- KTH: This dataset includes 25 individuals performing six different actions: walking, jogging, running, boxing, waving, and clapping. The complexity of human movements arises from the unique variations each individual displays while executing these actions. By examining previous frames, the model can understand the subtleties of human dynamics and predict future extended postural changes.
- SEVIR: This dataset consists of weather images that have been sampled and aligned using radar and satellite data. It is designed as a foundational resource to support algorithm development in meteorological research.
- DRS: This dataset describes the diffusion process of nonlinear wave, which satisfies the diffusion equation.
- FireSys: The FireSys dataset comprises data associated with fire observations, capturing both temporal and spatial trends of fire evolution, which faithfully represent the progression status in a natural setting.

## C Broader Impact

The development and application of the CaPaint framework in spatio-temporal (ST) dynamics bring several positive broader impacts. Understanding these impacts is crucial for responsible AI research and deployment.

**1. Data Imputation in Sparse Scenarios:** CaPaint excels in sparse data scenarios, effectively filling in missing data. This reduces the need for extensive sensor deployments, significantly lowering the cost associated with sensor installation. By optimizing data coverage and utilization, CaPaint not only enhances resource efficiency but also achieves substantial cost savings.

**2. Enhanced Predictive Accuracy and Interpretability:** CaPaint can identify and intervene in non-causal regions, improving the predictive accuracy and interpretability in various ST domains such as meteorology, human mobility, and disaster management. This improvement leads to better decision-making processes and resource allocation, ultimately benefiting society by providing more reliable and understandable predictive models.

**3. Cost-Effective Solutions:** By reducing the complexity of optimal ST causal discovery models, CaPaint offers a cost-effective solution for handling high-dimensional ST data. This makes advanced predictive technologies more accessible across a broader range of applications, particularly in fields with limited computational resources.

**4. Promotion of Causal Reasoning in AI:** The integration of causal reasoning into ST models encourages the development of AI systems that better mimic human understanding of cause-and-effect relationships. This can lead to more robust AI models capable of generalizing across different scenarios, fostering trust and reliability in AI applications.

**5. Innovation in Data Augmentation Techniques:** CaPaint introduces novel data augmentation methods using diffusion inpainting, which can inspire further research and innovation in data augmentation and ST prediction. This can lead to the emergence of new techniques, enhancing the robustness and performance of AI models in various domains.

The CaPaint framework represents a significant advancement in the field of ST dynamics, particularly in its ability to address sparse data scenarios, which reduces the need for extensive sensor deployments and lowers associated costs. Additionally, CaPaint enhances predictive accuracy, interpretability, and efficiency, promotes causal reasoning in AI, and introduces innovative data augmentation techniques. Responsible AI research and deployment should leverage these strengths to maximize benefits while minimizing risks.

## D    Metrics

In our research, we investigate the performance of our models using Mean Squared Error (MSE), Mean Absolute Error (MAE), and Structural Similarity Index Measure (SSIM). The formulas for evaluating these indicators, converted into decibels (dB) where applicable, are as follows:

**Mean Squared Error (MSE)**

Mean Squared Error (MSE) measures the average of the squares of the errors, that is, the average squared difference between the estimated values and the actual value. The MSE is given by:

$$\text{MSE} = \frac{1}{N} \sum_{i=1}^{N} (Y_i - \hat{Y}_i)^2 \tag{D.1}$$

where $Y_i$ is the actual value, $\hat{Y}_i$ is the predicted value, and $N$ is the number of observations.

**Mean Absolute Error (MAE)**

Mean Absolute Error (MAE) measures the average magnitude of the errors in a set of predictions, without considering their direction. It is the average over the test sample of the absolute differences between prediction and actual observation where all individual differences have equal weight. The MAE is given by:

$$\text{MAE} = \frac{1}{N} \sum_{i=1}^{N} \left| Y_i - \hat{Y}_i \right| \tag{D.2}$$

where $Y_i$ is the actual value, $\hat{Y}_i$ is the predicted value, and $N$ is the number of observations.

**Structural Similarity Index Measure (SSIM)**

Structural Similarity Index Measure (SSIM) is used for measuring the similarity between two images. The SSIM index is a decimal value between -1 and 1, where 1 is only reachable in the case of two identical sets of data. The SSIM formula can be quite complex due to its consideration of luminance, contrast, and structure comparison functions between the two images:

$$\text{SSIM}(x, y) = \frac{(2\mu_x \mu_y + C_1)(2\sigma_{xy} + C_2)}{(\mu_x^2 + \mu_y^2 + C_1)(\sigma_x^2 + \sigma_y^2 + C_2)} \tag{D.3}$$

where $\mu_x$, $\mu_y$ are the average of $x$ and $y$ respectively, $\sigma_x^2$, $\sigma_y^2$ are the variance of $x$ and $y$ respectively, $\sigma_{xy}$ is the covariance of $x$ and $y$, and $C_1, C_2$ are variables to stabilize the division with weak denominator.

## E    Limitations

While the implementation of the CaPaint method has demonstrated significant improvements in prediction accuracy and detail preservation in spatio-temporal forecasting tasks, its enhancements are most pronounced in scenarios characterized by data scarcity or uneven data distribution. In contexts where datasets are abundant and exhibit a broad and uniform distribution, the incremental gains offered by CaPaint may not be as substantial. Nevertheless, the method remains effective, providing consistent, albeit smaller, improvements across diverse data environments.

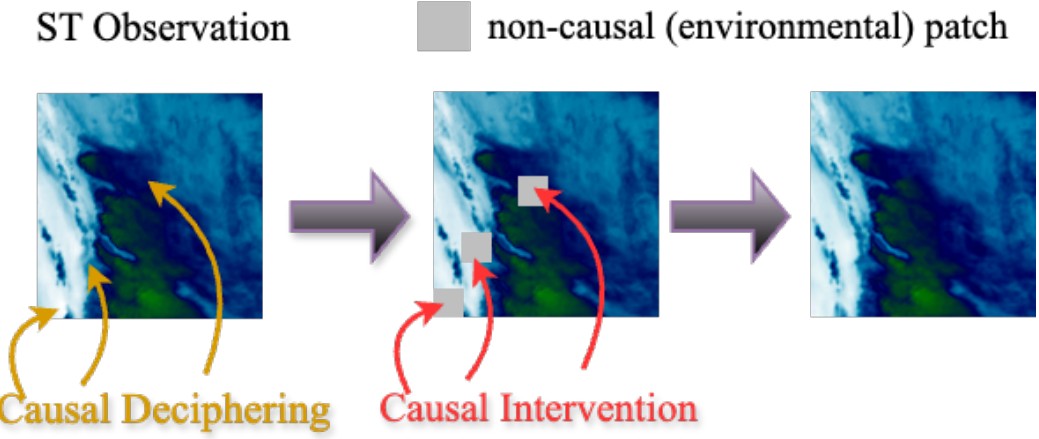

Figure F.1: Inpainting Example of our proposed CaPaint.

# F An example of ST Inpainting on SEVIR

The figure illustrates the process of maintaining causal regions intact while performing inpainting on non-causal (environmental) regions. The approach involves identifying and deciphering the causal regions (left), intervening by applying diffusion inpainting on the environmental patches (middle), and subsequently generating altered ST data copies (right). This method ensures that the intrinsic causal relationships within the data are preserved, while variations are introduced in the environmental context to augment the dataset effectively.

# G Uneven Distribution of Sensors Leading to Data Scarcity in Global Oceanic Observation

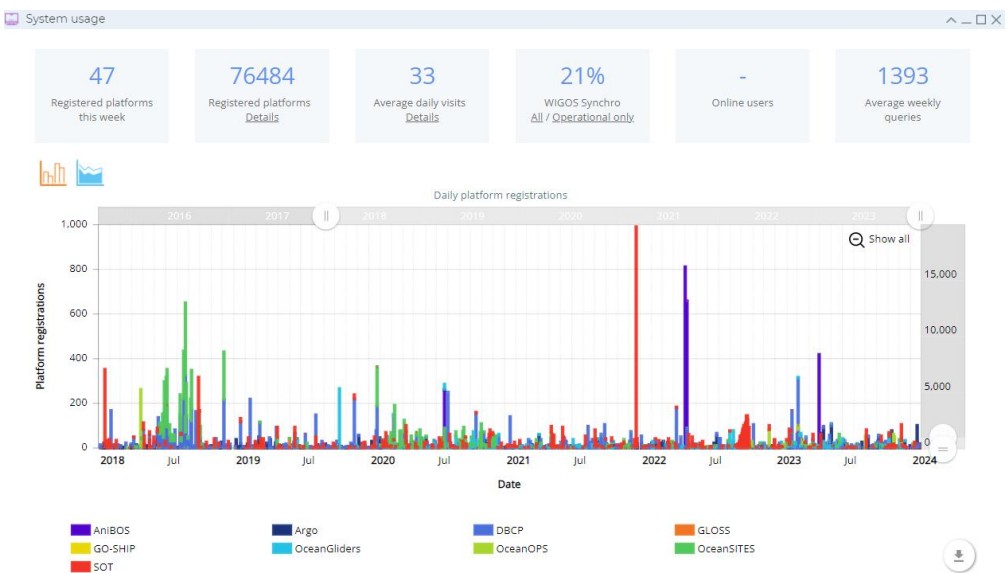

Figure G.1: Temporal distributional heterogeneity within the global oceanic observation platforms, which reveals that there are pronounced disparities in the deployment numbers of various types of sensors during different time intervals.

## H    Experimental Parameters

In this experiment, we employ different deep learning models and optimize them for training. All experiments are conducted on hardware equipped with 24 NVIDIA GeForce RTX 4090 GPUs. The optimizer used is Adam, and different learning rates (LR) and batch sizes are set for each model. The specific parameter settings are shown in the table below:

| Model | Learning Rate (LR) | Batch Size |
| --- | --- | --- |
| CLSTM | 0.001 | 8 |
| MAU | 0.001 | 8 |
| MMVP | 0.004 | 4 |
| PredRNNv2 | 0.001 | 8 |
| SimVP | 0.004 | 4 |
| ViT | 0.004 | 4 |
| Earthfarsser | 0.001 | 8 |

Table H.1: Learning rates and batch sizes for different backbones

These parameter settings are chosen based on the characteristics of each model and preliminary experimental results on the validation set, aiming to optimize the training efficiency and performance of the models. The Adam optimizer is used with a OneCycle learning rate scheduler, where the maximum learning rate is set according to the specified learning rate for each model, and the number of steps per epoch and the total number of epochs are set based on the training data and experimental setup. During the experiments, we ensure that all models are trained under the same hardware conditions to guarantee the comparability and reproducibility of the results.

## I    Visualizations on KTH

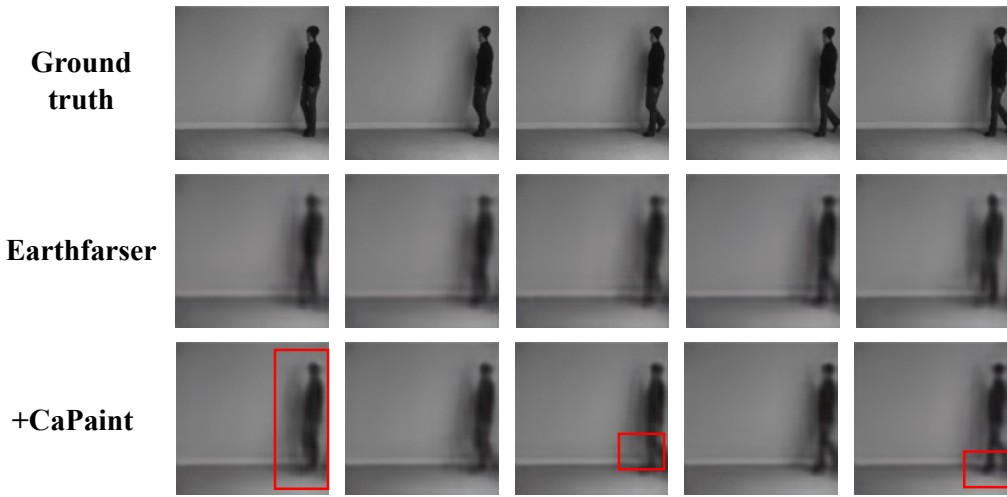

Figure I.1: Visualizations on KTH dataset showing the last 5 frames

The first row shows the ground truth for a walking individual. The second row, processed by Earthfarser, exhibits noticeable blurring and loss of detail. The third row, enhanced with +CaPaint, demonstrates a marked improvement in capturing fine details such as the shadow of the person and the accuracy of the foot motion, as highlighted in the red boxes.

## J    Visualizations on Diffusion Reaction System

The introduction of CaPaint has led to reductions in Mean Squared Error MSE and MAE, while the SSIM has shown improvements. These changes indicate that the CaPaint method effectively enhances model prediction accuracy and image quality. However, due to the high quality of the model predictions, the improvements might not be readily observable to the naked eye. Despite this, the

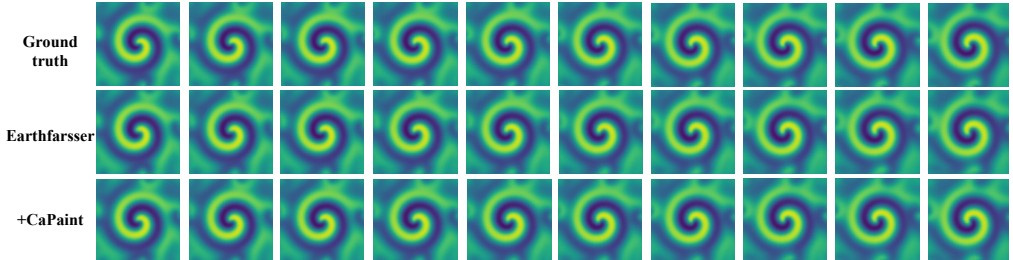
Figure J.1: Visualizations on DRS dataset showing 10 frames

positive effects of CaPaint are clearly evident through quantitative metrics, demonstrating its potential and practicality in enhancing the accuracy of complex dynamic systems predictions.

