# OpenReview forum: "Causal Deciphering and Inpainting in Spatio-Temporal Dynamics via Diffusion Model"
_NeurIPS.cc/2024/Conference — NeurIPS 2024 poster_

### Official Review · Reviewer_mbnW · 2024-06-15

**Soundness:** 4
**Presentation:** 3
**Contribution:** 4
**Rating:** 8
**Confidence:** 4

**Summary:**

The paper introduces CaPaint, a causal structure plugin for spatio-temporal (ST) forecasting, aiming to identify causal regions in data and enable the model to perform causal reasoning. Utilizing a two-stage process and employing a novel image inpainting technique using a fine-tuned unconditional Diffusion Probabilistic Model (DDPM), the paper proposes a method to fill in the gaps identified as environmental parts, enhancing model generalizability and interpretability significantly.

**Strengths:**

- The paper is well-written, with clear, concise explanations and the use of figures effectively illustrates the model's mechanisms and results.

- The paper introduces an interesting concept by incorporating causal inference into spatio-temporal data analysis, particularly through the integration of generative models

- The experiments are thoroughly conducted across multiple datasets and backbones, results are overall promising.

**Weaknesses:**

- The abstract contains a typo where front-door adjustment is incorrectly referred to as back-door adjustment.

- The paper appears to lack detailed descriptions on how the generated spatio-temporal data are synthesized into coherent ST sequences, missing crucial details on this aspect of the methodology.

- The paper does not clearly demonstrate how the quality and efficiency of generation are improved. It is recommended to supplement with additional experiments to substantiate these aspects.

**Questions:**

1. How effective is the inpainting technique implemented by CaPaint specifically on spatio-temporal datasets, and what are the key factors that influence its performance in these contexts?


2. Why do traditional data augmentation methods, which can disrupt spatio-temporal characteristics, result in performances that are consistent with or only slightly worse than the original, instead of showing a significant decline?


3. How does the performance compare when augmented data is combined with original data to form the training set for enhancing model generalizability, particularly when controlling for an equal amount of training data?

**Limitations:**

The authors highlight that the effectiveness of the method is limited under conditions of abundant data, as demonstrated through experiments that show more significant performance improvements under data-scarce conditions compared to when data is plentiful.

---

> ### Author Rebuttal · Authors · 2024-08-06
>
> **W1:** Thank you for your careful review. We appreciate your attention to detail and will correct this typo to accurately reflect the use of front-door adjustment in the abstract.
>
> **W2:** Thank you for your valuable feedback. For each original spatio-temporal sequence, we enhance the data by first identifying causal regions using a Vision Transformer. We then apply diffusion inpainting to fine-tune the data and fill the identified environmental parts. After inpainting, we perform sampling with a probability ppp to mix the original sequences with the augmented sequences. This process ensures that the generated data maintains spatial and temporal coherence, resulting in high-quality, coherent spatio-temporal sequences.
>
> **W3:** Thank you for your suggestion. We have supplemented our paper with additional experiments comparing our method with NuwaDynamics. The results demonstrate that our method achieves similar or better performance with only 2 sequences, whereas Nuwa requires 4-5 sequences. This clearly shows the superior quality and efficiency of our generation process.
>
> | Datasets    | SimVP   | SimVP | PredRNN-v2 | PredRNN-v2 |
> | ----------- | ------- | ----- | ---------- | ---------- |
> |             | CaPaint | Nuwa  | CaPaint    | Nuwa       |
> | **TaxiBJ+** | 2.21    | 2.56  | 2.86       | 3.25       |
> | **KTH**     | 31.26   | 33.98 | 38.45      | 40.37      |
>
> We selected MAE as the reference metric, and it is evident that our proposed CaPaint outperforms Nuwa across the board. It is worth mentioning that Nuwa requires generating 4 and 5 augmented sequences per spatio-temporal sequence for TaxiBJ+ and KTH, respectively, to achieve its results. In contrast, we only needed to generate 2 sequences to surpass Nuwa's performance. This is because Nuwa can only mix local information around the environment, whereas our method can consider and fill in global details, generating higher-quality data. This aligns with the background of our paper, which addresses data scarcity and uneven sensor collection.
>
>
>
> **Q1:** Thank you for your insightful question. We have provided an example of the inpainting technique in Appendix F. CaPaint effectively fills the environmental regions of the images based on their data distribution. We fine-tuned the Stable Diffusion model on our datasets, enabling it to learn the spatio-temporal data distributions accurately.
>
> The key factors influencing the performance of the inpainting technique in these contexts include:
>
> 1. **Data Distribution**: The model's ability to understand and replicate the inherent distribution of the spatio-temporal data.
> 2. **Model Fine-tuning**: The extent to which the Stable Diffusion model has been fine-tuned on the specific datasets to capture the nuances of the data.
> 3. **Environmental Region Identification**: The accuracy in identifying and masking the environmental patches that need inpainting.
> 4. **Causal and Non-causal Patch Distinction**: The precision in distinguishing between causal and non-causal patches to ensure that the inpainting enhances the model’s generalizability and interpretability
>
>
>
> **Q2:** Thank you for your question. We briefly analyzed this in Section 4.4 of our experiments. Traditional data augmentation methods indeed disrupt spatio-temporal characteristics, leading to two possible outcomes:
>
> 1. **Data Only Augmentation**: If only the data is augmented without corresponding changes to the labels, there is a significant decline in performance due to the loss of spatio-temporal coherence.
> 2. **Data and Label Augmentation**: If both the data and labels are augmented together, the performance only slightly declines. This is because augmenting both preserves some of the spatio-temporal characteristics, maintaining a level of coherence between the input data and the expected output.
>
> By retaining some of the intrinsic spatio-temporal properties through coordinated augmentation of both data and labels, the performance remains relatively stable.
>
>
>
> **Q3:** Thank you for your question. As shown in Figure 7 of our experiments section, even when maintaining an equal amount of training data, CaPaint demonstrates superior performance compared to the original backbone model. For instance, when comparing 50% original data with 25% original data combined with 25% augmented data, CaPaint's MAE and MSE are lower than those of the original backbone. This indicates that CaPaint effectively enhances model generalizability by leveraging the augmented data.
>
> | Metric        | 10%    | 25%    | 50%    | 75%    | 100%   |
> | ------------- | ------ | ------ | ------ | ------ | ------ |
> | **SimVP**     |        |        |        |        |        |
> | MAE           | 0.4925 | 0.4477 | 0.3215 | 0.2438 | 0.2320 |
> | MSE           | 0.5103 | 0.4434 | 0.2821 | 0.1843 | 0.1645 |
> | **SimVP+CaP** |        |        |        |        |        |
> | MAE           | 0.3925 | 0.2875 | 0.2633 | 0.2210 | 0.2057 |
> | MSE           | 0.3787 | 0.2541 | 0.2157 | 0.1586 | 0.1390 |
>
> I believe my response has addressed your concerns. If you have any further questions, please feel free to let me know. Thank you!

---

> > ### Comment · Reviewer_mbnW · 2024-08-09
> >
> > Dear Authors,
> >
> > Thank you for your detailed and clear response to the issues we raised. After careful review, we find that your replies are very specific and have adequately clarified the concerns mentioned in our review. In particular, your detailed experimental results and further explanations of the methodology have provided us with a more comprehensive understanding of the paper. Additionally, your selection of error metrics and discussion of different data augmentation methods have deepened our appreciation of the model you proposed. We believe that your response has effectively addressed all the questions we previously raised, and the additional experimental results further enhance the validity and robustness of your method. Moreover, we especially appreciate the innovative application of diffusion inpainting to spatio-temporal video data, which significantly improves the model's performance. Based on these improvements and supplements, we will raise our score for your paper.
> >
> > Thank you again for your careful attention and detailed responses.
> >
> > Best regards,
> > Reviewer

---

> > > ### Author Response · Authors · 2024-08-09
> > >
> > > Dear Reviewer,
> > >
> > > Thank you very much for your thoughtful and positive feedback. We greatly appreciate your recognition of our efforts to address the concerns raised in the initial review. We are delighted that our detailed explanations, additional experimental results, and the innovative application of diffusion inpainting have enhanced your understanding and appreciation of our work.
> > >
> > > Your acknowledgment of the improvements we've made means a great deal to us, and we are grateful for your willingness to raise the score based on these enhancements. We will continue to refine and improve our research to contribute to the field.
> > >
> > > Thank you again for your careful consideration and support.
> > >
> > > Best regards

---

### Official Review · Reviewer_nGNj · 2024-07-09

**Soundness:** 3
**Presentation:** 3
**Contribution:** 3
**Rating:** 6
**Confidence:** 2

**Summary:**

The paper focuses on generalizability and interpretability for spatio-temporal predicting. The authors propose a causal structure plugin, named CaPaint, which identifies causal regions in data to generate data for scenarios where data are scarce. Experiments on five datasets demonstrate the effectiveness of the proposed method in spatio-temporal forecasting.

**Strengths:**

1.The paper focuses on the issue of modeling uneven and insufficient spatio-temporal data, which is a fascinating and significant area of research.

2.To incorporate physical laws into deep networks, the authors propose a method that obeys the causal deciphering and performs interventions on the non-causal diffusion pathces, which is an extremely challenging problem.

3.The authors have conducted experiments on five datasets, validating the effectiveness of the model, and have appropriately discussed the limitations of the model.

**Weaknesses:**

1.In line 39-44, the authors lack discussion of why the causality and interpretability of models can improve generalization capabilities when dealing with the uneven, insufficient data collection.

2.In the left side of Fig.4, the visualizations of finer details are small and not clear enough, so it would be more informative to zoom in on local details of the image.

**Questions:**

Please see the Weaknesses.

**Limitations:**

The authors adequately addressed the limitations.

---

> ### Author Rebuttal · Authors · 2024-08-02
>
> **Q1:** In line 39-44, the authors lack discussion of why the causality and interpretability of models can improve generalization capabilities when dealing with the uneven, insufficient data collection.
>
> **Answer:**
>
> 1. When data is sparse, models may learn **shortcut solutions from biased data**, as noted in several studies [1-3]. These shortcuts can significantly interfere with causal discovery, which relies on counterfactual reasoning to understand the underlying mechanisms that generate the observed data. In spatio-temporal domains, this issue is particularly critical as the temporal dependencies and spatial correlations are more complex and often underexplored. Traditional approaches have primarily focused on **spatio-temporal graphs**, but our work extends this to spatio-temporal continuity in **image data (video data)**, a relatively underexplored area. By addressing these gaps, we provide a more robust foundation for understanding and predicting spatio-temporal dynamics, ultimately enhancing the model's ability to generalize from incomplete or biased datasets.
> 2. **Interpretability and causality** are strongly correlated. Discovering patterns in model and data predictions is crucial for identifying causality and enhancing interpretability. This process allows us to understand how different variables interact and contribute to the observed outcomes. In our work, we **leverage self-supervision** and employ global perception by **introducing inpainting to dynamically fill in potentially biased areas.** This approach ensures that the model can make informed predictions based on a more comprehensive understanding of the underlying data distribution, thus achieving interpretability. This interpretability, grounded in solid causal theory, allows us to make sense of the model's decisions, facilitating better generalization and robustness, especially in scenarios with uneven and insufficient data collection.
> 3. To help readers better understand the relationship between data scarcity, causality, and interpretability, we have **cited more relevant literature** in the paper. These citations not only provide a historical context for the technical developments in this area but also highlight the critical correlation between data scarcity, causality, and interpretability. By grounding our discussion in established research, we illustrate how our approach aligns with and advances current understanding, providing a clearer picture of how addressing **causality and interpretability** can significantly improve generalization capabilities in spatio-temporal data analysis.
>
> **Q2:** In the left side of Fig.4, the visualizations of finer details are small and not clear enough, so it would be more informative to zoom in on local details of the image.
>
> **Answer:** Thank you for pointing this out. We appreciate your feedback and recognize the importance of clear and detailed visualizations. To address this issue, we will provide zoomed-in versions of the local details for different datasets to offer a clearer and more informative visualization. To improve the readability of the paper, we have re-arranged the layout of Figure 4. We have enlarged the relevant images and fonts to make it easier for readers to view the finer details. Additionally, we have systematically refined the layout of the entire paper, which includes adjustments to image sizes, fonts, and tables to enhance overall readability**.Upon acceptance, we will include these additional results and enhanced visualizations in the appendix for ease of reference.** This will ensure that all visual details are accessible and that the readers can fully appreciate the improvements and performance of our proposed method.
>
> I believe my response has addressed your concerns. If you have any further questions, please feel free to let me know. Thank you!
>
> **Reference**:
>
> [1] Causal Attention for Interpretable and Generalizable Graph Classification.
>
> [2] Enhancing Out-of-distribution Generalization on Graphs via Causal Attention Learning.
>
> [3] Reinforced causal explainer for graph neural networks.

---

> > ### Comment · Reviewer_nGNj · 2024-08-13
> >
> > Thank you for the detailed explanation and the improved visualization. Their discussion effectively highlights the advantages of their approach, offering a thorough understanding of how causality and interpretability contribute to improved generalization. In addition, the changes on visualitzations will enhance the readability and comprehension of the paper. While the authors have addressed the issues raised in a satisfactory manner, we have decided to withhold our score at this time. We appreciate the effort put into the rebuttal and look forward to seeing the final revisions.

---

> > > ### Author Response · Authors · 2024-08-13
> > >
> > > Dear Reviewer,
> > >
> > > Thank you very much for your considerate feedback and for acknowledging our efforts in addressing the issues raised. We appreciate your kind words regarding the improvements and the detailed explanations provided. Your insights have been invaluable in guiding our revisions, and we are committed to making the final adjustments to further enhance the quality of our paper.
> > >
> > > Thank you again for your support and thoughtful evaluation.
> > >
> > > Best regards,

---

### Official Review · Reviewer_s4LS · 2024-07-09

**Soundness:** 2
**Presentation:** 2
**Contribution:** 2
**Rating:** 5
**Confidence:** 3

**Summary:**

The paper presents CaPaint to improve spatio-temporal predictions by identifying causal regions and employing diffusion inpainting techniques. The approach addresses the challenges of high computational costs in ST causal discovery.

**Strengths:**

1.  CaPaint seamlessly integrates with a variety of existing spatio-temporal prediction models. The paper thoroughly evaluates the method using diverse backbone models, showcasing the robustness and versatility of CaPaint across different scenarios.
2. The experimental results across five real-world ST benchmarks demonstrate substantial improvements
3. The combination of causal inference and diffusion models is sound.

**Weaknesses:**

1. The novelty of the proposed method is somewhat limited. The concept of causal patch discovery was already introduced in NuwaDynamics. This work primarily builds on that by utilizing diffusion models for data generation and proposing a different SCM, which is not necessarily better.
2. The evaluations in Table 1 use the same datasets as NuwaDynamics. To ensure a fair comparison and better highlight the improvements, it is recommended that the authors use the same settings as NuwaDynamics and directly compare their results with it.

**Questions:**

see weakness

**Limitations:**

The technical contribution of this work is somewhat incremental, providing only limited improvements compared to NuwaDynamics.

---

> ### Author Rebuttal · Authors · 2024-08-06
>
> **Answer(Q1):** Thank you for your feedback. We **respectfully** disagree with the assessment that our method lacks novelty. Our work presents significant improvements and innovations over NuwaDynamics.
>
> 1. **Conceptual Difference**:
>    - Our proposed method employs a different approach to causal adjustments, utilizing front-door adjustment rather than back-door adjustment as in NuwaDynamics. Back-door adjustment requires traversing and repeatedly sampling all environmental patches, leading to exponential complexity: ${\cal O}( {T \times {\cal N}_E^{{\cal M}\left( {\rm{*}} \right)}} )$
>    - In contrast, our method with front-door adjustment avoids the need for repeated sampling of all environmental patches, reducing the complexity to linear levels: $\cal{O}(T \times {\cal N}_E)$
>    - This significant reduction in complexity provides practical advantages, especially in large-scale data processing and real-time applications, by substantially lowering computational resource requirements.
> 2. **Different Methodology**:
>    - Our approach involves the use of generative models, specifically diffusion inpainting methods, for causal interventions. After fine-tuning the model, it achieves a global awareness of the data, enabling effective inpainting of environmental regions. In contrast, NuwaDynamics can only perceive local information around the surrounding areas.
>    - Traditionally, Diffusion Models (DMs) were utilized to generate high-quality synthetic images in the field of Computer Vision (CV). Inspired by the powerful generation capability of DMs given an input picture, we incorporate generative models like DMs into spatio-temporal video data analysis for data augmentation, without harming its efficiency greatly. Our key insight lies in addressing the challenge of data sparsity, and the experimental results have demonstrated that our method achieves significant improvements. This is a substantial innovation in this field.
> 3. **Broad Applicability**:
>    - Our CaPaint method effectively addresses the issue of exponential complexity found in NuwaDynamics, resulting in more efficient processing. Furthermore, the spatio-temporal data sequences generated by our method are of higher quality, enhancing the robustness and generalizability of the model. This allows CaPaint to be applied in various scenarios, making it advantageous for practical applications and deployment in the industry.
>
> By addressing these aspects, we hope to clarify the distinct advantages and innovations of our method compared to existing approaches.
>
>
>
> **Answer(Q2):** Thank you for your suggestion. Our CaPaint and Nuwa approaches inherently differ in their focus on front-door and back-door adjustments, respectively. Back-door adjustment, as used in Nuwa, requires traversing and repeatedly sampling as many environmental patches as possible, which leads to fundamentally different experimental setups. In contrast, front-door adjustment, which we employ, only needs to perceive the overall environment and fill in the environmental parts. This eliminates the need for repeatedly sampling numerous environmental patches, which is one of the key advantages of our method.
>
> Additionally, in the experimental section of our paper, we have provided a comparison between our method and Nuwa under our settings. The results show that our method outperforms Nuwa across different datasets. This demonstrates the robustness and effectiveness of our approach.
>
> | Datasets    | Flip  | Rotate | Crop  | NuWa  | CaPaint |
> | ----------- | ----- | ------ | ----- | ----- | ------- |
> | **DRS**     | 2.10  | 2.11   | 2.34  | 2.02  | 1.57    |
> | **KTH**     | 23.15 | 23.14  | 23.11 | 22.32 | 20.56   |
> | **SEVIR**   | 15.41 | 15.45  | 15.95 | 15.14 | 14.63   |
> | **TaxiBJ+** | 16.47 | 16.39  | 15.94 | 15.11 | 12.87   |
> | **FireSys** | 17.02 | 17.07  | 17.15 | 16.68 | 15.79   |
>
>
>
> To better explore the results of our method and Nuwa under the same settings, we followed your suggestion to align the experimental setup with that of Nuwa. Additionally, our experiments introduced backbones that have been updated in recent years, meaning the backbones we used are newer than those in Nuwa. Due to limited time and resources, we selected overlapping backbones such as  SimVP and PredRNN-V2, and overlapping datasets,TaxiBJ+ and KTH for comparison, setting the environmental ratio to 15% for testing.
>
> | Datasets    | SimVP   | SimVP | PredRNN-v2 | PredRNN-v2 |
> | ----------- | ------- | ----- | ---------- | ---------- |
> |             | CaPaint | Nuwa  | CaPaint    | Nuwa       |
> | **TaxiBJ+** | 2.21    | 2.56  | 2.86       | 3.25       |
> | **KTH**     | 31.26   | 33.98 | 38.45      | 40.37      |
>
> We selected MAE as the reference metric, and it is evident that our proposed CaPaint outperforms Nuwa across the board. It is worth mentioning that Nuwa requires generating 4 and 5 augmented sequences per spatio-temporal sequence for TaxiBJ+ and KTH, respectively, to achieve its results. In contrast, we only needed to generate 2 sequences to surpass Nuwa's performance. This is because Nuwa can only mix local information around the environment, whereas our method can consider and fill in global details, generating higher-quality data. This aligns with the background of our paper, which addresses data scarcity and uneven sensor collection.
>
> In summary, our more advanced concept is fundamentally different from Nuwa and is more advantageous for practical applications. I believe my response has addressed your concerns. If you have any further questions, please feel free to let me know. Thank you!

---

> > ### Comment · Reviewer_s4LS · 2024-08-11
> >
> > Thanks for the authors' rebuttal and additional experimental results. I recommend the authors to add the comparison results with Nuwa to the main paper. I have raised my rating accordingly.

---

> > > ### Author Response · Authors · 2024-08-11
> > >
> > > Dear Reviewer,
> > >
> > > Thank you for your thoughtful feedback. We sincerely appreciate your constructive suggestions and are grateful for your support throughout this process.
> > >
> > > Best regards,
> > > Authors

---

### Official Review · Reviewer_botx · 2024-07-13

**Soundness:** 3
**Presentation:** 3
**Contribution:** 3
**Rating:** 6
**Confidence:** 2

**Summary:**

This paper introduces a groundbreaking framework named CaPaint, which is designed to tackle the critical issues of data scarcity and the absence of causal connections in spatiotemporal (ST) prediction models. The authors have established a robust causal framework that not only identifies regions within data that exhibit causal relationships but also endows the model with the capability to reason about causality during a two-stage processing procedure. In the initial stage, they leverage self-supervised Vision Transformer (ViT) reconstruction to identify the crucial causal patches within ST observations. This is followed by an intervention phase where they employ diffusion inpainting techniques to manipulate non-causal areas while preserving the integrity of core causal areas. The innovative method reduces the complexity of generating data from exponential levels to quasi-linear levels, thereby significantly enhancing efficiency. Moreover, it has shown remarkable improvements across various ST benchmarks by integrating diffusion models as a novel data augmentation technique, marking a paradigm shift for this field.

**Strengths:**

- Addresses data scarcity and lack of causal connections in ST prediction models effectively.
- Novel Method: Innovative use of self-supervised Vision Transformer reconstruction for causal patch identification. And employs diffusion inpainting techniques to manipulate non-causal areas, preserving core causal integrity.
- Demonstrates significant improvements across various ST benchmarks, integrating diffusion models as a novel data augmentation technique. Besides, it reduces data generation complexity from exponential to a quasi-linear level.

**Weaknesses:**

- Details on computational efficiency or scalability of the proposed method are not provided, leaving it as a potential limitation for practical applications.
- More visualization of the prediction results should be included even in supplementary material.

**Questions:**

Mentioned in the weakness section.

**Limitations:**

I think the author addressed the limitation mentioned in this work.

---

> ### Author Rebuttal · Authors · 2024-08-02
>
> - Details on computational efficiency or scalability of the proposed method are not provided, leaving it as a potential limitation for practical applications.
>
> Thank you for your valuable feedback. We appreciate your concern regarding the computational efficiency and scalability of our proposed method.
>
> **1:** We leverage the attention mechanism for causal deciphering, which does not introduce additional parameters. This ensures that our model remains efficient without incurring extra computational overhead. **2:** In the Introduction, we briefly analyze the computational efficiency of our method. Specifically, traditional backdoor adjustment requires traversing and repeatedly sampling all environmental patches, leading to exponential complexity: ${\cal O}( {T \times {\cal N}_E^{{\cal M}\left( {\rm{*}} \right)}} )$. In contrast, our CaPaint method employs front-door adjustment, which eliminates the need for repeated sampling of all environmental patches. This improvement reduces the complexity to linear levels: $\cal{O}(T \times {\cal N}_E)$.
>
> **Reduction in Complexity**: Our approach substantially reduces the complexity of the optimal spatio-temporal causal discovery process from exponential to quasi-linear levels. By performing targeted interventions only on identified environmental patches rather than the entire dataset, we achieve significant computational savings.
>
> **Scalability**: This reduction in complexity ensures efficient causal interventions, making our method more practical and scalable for real-world applications. Our method has been tested across multiple benchmark datasets (FireSys, SEVIR, Diffusion Reaction System (DRS), KTH, and TaxiBJ+), demonstrating its robustness and effectiveness in enhancing model performance.
>
> **Practical Applications**: By implementing these enhancements, CaPaint provides a robust framework for spatio-temporal dynamics with improved computational efficiency and scalability. This makes it suitable for deployment in various real-world scenarios where data scarcity and computational efficiency are critical concerns.
>
> - More visualization of the prediction results should be included even in supplementary material.
>
> Thank you for your suggestion. In the main body of the paper, we have presented the visualization results for the SEVIR and TaxiBJ+ datasets. Additionally, we have included the visualization results for the KTH and DRS datasets in Appendix I and J, respectively. Although these visualizations are not in the supplementary material section, they are provided in the appendix. Furthermore, Appendix F also showcases the visualization of the inpainting effects of our CaPaint method on the SEVIR dataset.While we believe these visualizations adequately demonstrate the performance advantages of incorporating CaPaint, we are continuously updating our results to include more comprehensive visualizations. Specifically, we plan to add:
>
> 1. **Visualizations for More Datasets**: We will supplement the visualizations with results from the FireSys dataset.
> 2. **Visualizations for Different Backbones**: We will include visualization results using different backbone models.
> 3. **Comparisons with Different Augmentation Methods**: We will provide comparisons of visualizations using various augmentation methods.
>
> Upon acceptance, we will include these additional results in the appendix for ease of reference. These updates will ensure that our paper provides a comprehensive overview of the performance improvements achieved by our method and its applicability to a wide range of datasets and model architectures.
>
> I believe my response has addressed your concerns. If you have any further questions, please feel free to let me know. Thank you!

---

### Author Rebuttal · Authors · 2024-08-06

Dear Reviewers,

We would like to extend our sincere gratitude to all reviewers for their thorough and insightful feedback on our manuscript. We appreciate the time and effort you have invested in evaluating our work. Below, we provide an overall summary addressing the common strengths, weaknesses, and suggestions highlighted in your reviews.

#### Common Strengths:

1. **Addressing Data Scarcity and Causal Connections**:  (Reviewer `botx`, `nGNj`,`mbnW`)
   - Multiple reviewers noted that our work effectively addresses the critical issues of data scarcity and the lack of causal connections in spatio-temporal (ST) prediction models. We are pleased that our efforts to establish a robust causal framework were recognized.
2. **Innovative Methodology**: (Reviewer `botx`, `s4LS`,`mbnW`)
   - The innovative use of self-supervised Vision Transformer (ViT) reconstruction for causal patch identification and the employment of diffusion inpainting techniques were highlighted as significant strengths. We are encouraged that our novel approach was well-received.
3. **Performance Improvements**: (Reviewer `botx`, `nGNj`,`mbnW`, `s4LS`)
   - Reviewers acknowledged the substantial improvements demonstrated across various ST benchmarks, validating the robustness and versatility of CaPaint. The integration of diffusion models as a novel data augmentation technique was noted as a key contribution.
4. **Soundness and Contribution**:(Reviewer `botx`, `nGNj`,`mbnW`, `s4LS`,)
   - The overall soundness and contribution of our work were rated positively. We are grateful for the recognition of the technical rigor and the potential impact of our research in advancing the field of spatio-temporal data analysis.

#### Common Weaknesses and Our Responses:

1. **Computational Efficiency and Scalability**: (Reviewer `botx`, `mbnW`)
   - Several reviewers expressed concerns regarding the computational efficiency and scalability of our method. We have addressed this by elaborating on how our use of the attention mechanism and front-door adjustment reduces complexity from exponential to quasi-linear levels, ensuring efficient causal interventions suitable for real-world applications.
2. **Visualization of Prediction Results**: (Reviewer `botx`, `nGNj`)
   - Reviewers suggested including more visualizations of prediction results, even in supplementary material. We have updated our paper to include zoomed-in versions of local details for different datasets, re-arranged the layout for better readability, and added more comprehensive visualizations for additional datasets, backbones, and augmentation methods. These enhancements will be included in the appendix upon acceptance.
3. **Clarification on Causality and Interpretability**: (Reviewer `mbnW`, `s4LS`,)
   - The relationship between causality, interpretability, and their impact on generalization capabilities when dealing with uneven and insufficient data collection was pointed out as lacking. We have enriched our discussion by citing more relevant literature and providing a detailed explanation of how our approach leverages self-supervision and inpainting to dynamically address biased areas, thereby achieving robust causal discovery and interpretability.
4. **Comparison with NuwaDynamics**: (Reviewer `s4LS`,)
   - Reviewers noted the need for a more direct comparison with NuwaDynamics using the same settings. We have aligned our experimental setup with NuwaDynamics and provided additional experiments to highlight our method's superior performance with fewer sequences, demonstrating the efficiency and quality of CaPaint.

#### Additional Improvements:

- **Typographical Corrections**: We have corrected the typo in the abstract where front-door adjustment was incorrectly referred to as back-door adjustment.
- **Detailed Methodological Clarifications**: We have provided more detailed descriptions on the synthesis of spatio-temporal data into coherent sequences and included additional experiments to substantiate the improvements in quality and efficiency of generation.

In conclusion, we are grateful for the positive feedback and constructive criticisms provided by the reviewers. Your comments have been invaluable in helping us improve our manuscript. We believe that the revisions and additional experiments we have incorporated address the concerns raised and enhance the overall quality and clarity of our work.

Thank you once again for your detailed and thoughtful reviews. We look forward to your continued feedback and hope our revised submission meets your expectations.

Warm regards,

Authors

---

### Decision · Program_Chairs · 2024-09-25

**Decision:**

Accept (poster)

**Comment:**

We thank the authors for their submission.  Reviewers agreed that the work is a novel and thoroughly examined way to incorporate causal assumptions into spatiotemporal models and imputation.  Reviewers also appreciated the additional experimental results directly comparing the proposed CaPaint and Nuwa methods.